# An Overlooked Bone Metabolic Disorder: Cigarette Smoking-Induced Osteoporosis

**DOI:** 10.3390/genes13050806

**Published:** 2022-04-30

**Authors:** Weidong Weng, Hongming Li, Sheng Zhu

**Affiliations:** 1Department of Trauma and Reconstructive Surgery, BG Trauma Clinic, Siegfried Weller Institute for Trauma Research, Eberhard Karls University Tuebingen, 72076 Tuebingen, Germany; wengweidong5657@gmail.com; 2Department of Orthopedics, Xiangya Hospital, Central South University, Changsha 410008, China; hominglee0309@163.com

**Keywords:** cigarette smoking, osteoporosis, RANK-RANKL-OPG pathway, AhR pathway

## Abstract

Cigarette smoking (CS) leads to significant bone loss, which is recognized as an independent risk factor for osteoporosis. The number of smokers is continuously increasing due to the addictive nature of smoking. Therefore it is of great value to effectively prevent CS-induced osteoporosis. However, there are currently no effective interventions to specifically counteract CS-induced osteoporosis, owing to the fact that the specific mechanisms by which CS affects bone metabolism are still elusive. This review summarizes the latest research findings of important pathways between CS exposure and bone metabolism, with the aim of providing new targets and ideas for the prevention of CS-induced osteoporosis, as well as providing theoretical directions for further research in the future.

## 1. Introduction

Osteoporosis is a musculoskeletal disorder characterized by bone remodeling imbalance, contributing to a high risk of fragility fractures [1]. Osteoporosis and osteoporosis-related fractures are recognized as a worldwide public health challenge [2]. According to the literature survey, there are over 200 million individuals with osteoporosis [3]. Patients with osteoporosis being older than 50 are closely correlated with an increased risk of osteoporotic fractures, particularly in postmenopausal women [4,5]. Additionally, it should be noted that treatment for osteoporosis and osteoporotic fractures is costly, with an estimated annual cost of EUR 55 billion to the European Union [6].

Tobacco consumption represents one of the most common risk factors affecting the progression of diseases across the world. As reported by the World Health Organization, tobacco consumption is responsible for nearly 6.5 million deaths per year throughout the world [7]. It estimates that there will be approximately 8 million deaths per year attributed to tobacco consumption throughout the world by 2050 [7]. Additionally, cigarette smoking (CS) is the most commonly accepted approach to consuming tobacco. Based on their smoking status, smokers are divided into four categories: non-smokers (<100 cigarettes throughout their life), light smokers (1–10 cigarettes per day), moderate smokers (11–19 cigarettes per day) and heavy smokers (≥20 cigarettes per day) [8].

Cigarette smoking (CS) is recognized as an independent risk factor for the development of osteoporosis. Clinical studies have illustrated that smokers have significantly lower bone mineral density (BMD) than non-smokers (Table 1), and cumulative bone loss can increase their lifetime risk of hip fracture by 50% [9,10]. It has been shown that long-term CS can lead to an imbalance of bone turnover, further contributing to the reduction in bone mass and bone length and increased risk of fractures [11,12,13,14]. Furthermore, chronic consumption of cigarettes has been increasingly linked to impaired muscle function [15,16].

Smoking cessation is the best strategy to attenuate the detrimental influence of CS on bone metabolism [17]. However, this strategy may not be reliable in smokers owing to withdrawal symptoms and the lack of smoking rituals during cessation [18]. There have been several attempts to replace cigarettes, such as nicotine chewing gums, transdermal patches and nasal sprays [19]. However, these alternatives have limited efficiency [18,19], as the number of smokers worldwide continues to increase. According to the latest survey, the fact that 1.1–1.3 billion people worldwide smoke, which can cause 8 million deaths per year [20], indicating that smoking addiction is extremely high, and cigarette control is not that effective.

Therefore, it is essential to find effective interventions for CS-induced osteoporosis. However, the specific effects and molecular mechanisms of smoking on bone metabolism and the development of osteoporosis are currently unclear [21]. In this review, we aim to summarize the key findings of CS on bone metabolism and update potential directions for the treatment of smoking-related osteoporosis.

**Table 1 genes-13-00806-t001:** Clinical studies of cigarette smoking in relation to osteoporosis.

Reference	Gender	Smoking Status	Population (n)	Age (Years Old)	Site	BMD Values (g/cm^2^)	*p* Value
Hollenbach et al. [22]	Male	Non-smokers	417	60–99	Hip	0.935 ± 0.008	<0.05
		Smokers	87	60–89		0.895 ± 0.016	
	Female	Non-Smokers	573	60–99		0.780 ± 0.005	<0.01
		Smokers	181	60–89		0.741 ± 0.010	
Marques et al. [23]	55.9% Female	Never-Smokers	1275	75.1 ± 4.7	Trabecular	0.1428 ± 0.00344	0.152
		Former-Smokers	1176	74.5 ± 4.7		0.1444 ± 0.00335	
		Current-Smokers	222	73.0 ± 4.6		0.01398 ± 0.00352	
Egger et al. [24]	Male	Never Smokers	42	63–67	Lumber	1.12	<0.05
		Ex-Smokers	140	63–69		1.07	
		Current Smokers	42	63–68		1.04	
	Female	Never Smokers	99	64–67		0.97	ns
		Ex-Smokers	64	63–67		0.90	
		Current Smokers	23	63–68		0.89	
Trevisan et al. [25]	Female	Never Smokers	812	66 ± 10	Lumber	0.77 ± 0.11	ns
		Former-Smokers	156	65 ± 10		0.78 ± 0.13	
		Current Smokers	99	61 ± 10		0.76 ± 0.11	
Bjarnason et al. [26]	Female	Non-Smokers	192	53.5 ± 1.9	Hip	0.86 ± 0.09	ns
		Smokers	78	53.1 ± 1.6		0.85 ± 0.1	

## 2. The Role of CS on Bone Metabolism: Pathophysiological Mechanisms

Due to the complex composition of cigarettes and the lack of standardized in vitro and in vivo experimental set-ups (Table 2), the pathophysiological mechanisms of CS on bone metabolism are still elusive. It is attributed to the absence of amounts of reliable and valid research to clearly define these mechanisms, and some observations have been somewhat debated [27]. Recent studies have presented mechanisms of the influences of CS on bone metabolism, including indirect and direct mechanisms [11,12,27,28].

### 2.1. Indirect Mechanisms

#### 2.1.1. Body Weight

It has been shown that body weight is negatively associated with long-term CS [34,35,36]. This is probably because nicotine, a major addictive substance in cigarettes, inhibits food ingestion by stimulating the secretion of both dopamine and serotonin [37]. Additionally, CS has been shown to enhance lipid oxidation, contributing to the reduction in body weight [38]. In 2007, Wong et al. provided the underlying mechanisms explaining why long-term CS resulted in a reduction in body weight: (i) weight loss results in a reduction in the mechanical loading on bone, contributing to a decreased stimulus for osteogenic differentiation; (ii) CS cause a reduction in adipose tissue by inhibiting the conversation of androgen to estrogen; and (iii) CS cause a reduction in leptin levels, which has been known to enhance bone mass [12].

#### 2.1.2. Parathyroid Hormone- (PTH-) Vitamin D Axis

The PTH-vitamin D axis exerts crucial functions in maintaining bone density and calcium hemostasis [39]. PTH has a regulatory function in regulating serum calcium levels by bone and kidney reabsorption, while vitamin D can modulate the absorption of calcium from the intestine [40,41]. Several studies have indicated that smoking downregulates vitamin D serum levels [42,43] and inhibits the production of PTH [44]. However, the influence of CS on the inhibition of the production of PTH remains controversial [45,46]. These inconsistencies may be due to confounding effects of body weight, alcohol consumption, medication use and calcium and vitamin D supplementation [42,47].

#### 2.1.3. Gonadal Hormones

Estrogen and testosterone exert essential functions in modulating bone remodeling [48,49]. It has been shown that estrogen inhibits osteoclast differentiation and bone resorption [50], while testosterone promotes proliferation and osteogenic differentiation [51]. In females, CS causes a reduction in estrogen levels by altering estrogen metabolism [52,53]. Several explanations behind this phenomenon are as follows: (i) nicotine and cotinine suppress aromatase enzyme activity, contributing to a reduction in estrogen levels [54]; (ii) CS promotes estradiol to decompose into 2-methoxyestrone in the liver [55]; and (iii) CS enhances serum hormone-binding globulin levels, leading to a reduction in free estradiol levels in the blood [56,57]. In males, current findings are contradictory. Some research indicated testosterone levels were comparable between smokers and nonsmokers, whereas other research showed that smokers had higher testosterone levels than nonsmokers [58,59]. Like females, the potential mechanism of aromatase inhibition was involved in males [44].

#### 2.1.4. Oxidative Stress

The presence of reactive oxygen species (ROS) in bone can influence resident cell behavior, extra-cellular matrix composition and tissue architecture. CS is related to high levels of ROS, which improves osteoclast activity and suppresses osteoblast activity, leading to a reduction in bone mass [60,61], and maqui berry extract, a fruit extract rich in anthocyanins, is effective against the effects of CSE on osteoblasts. A clinical study demonstrated that smoking increased levels of oxidative stress [62]. Consistently, Zhu and colleagues observed that cigarette smoke extract (CSE) significantly enhanced ROS and nitrotyrosine levels in human osteoblasts [63]. Probing potential molecular mechanisms, Aspera-Werz et al.reported that Nrf2 directly modulates antioxidant defense systems in human mesenchymal stem cells exposed to CSE [64], and resveratrol could decrease ROS level generated by CSE to protect the primary cilia integrity in human bone mesenchymal stem cells.

### 2.2. Direct Mechanisms

CS has been found to exert a direct role in bone tissue through binding to several receptors, including nicotinic acetylcholine receptors and androgen receptors in osteoblasts and aryl hydrocarbon receptors in osteoclasts [11]. Nicotine, a major addictive substance in cigarettes, binds to nicotinic receptors in osteoblasts [65]. However, the role of nicotine in bone metabolism has been controversial. Marinucci et al. observed that nicotine negatively contributed to the impaired osteogenic differentiation of mesenchymal stem cells (MSCs) [66]. On the contrary, Daffner and colleagues demonstrated that nicotine could even promote MSC proliferation and osteogenic differentiation at concentrations lower than its serum concentration in smokers [67]. Aspera-Werz et al. showed that the osteogenic differentiation of MSCs was not largely altered when the cells were exposed to nicotine at concentrations similar to its serum levels in smokers [64]. Similarly, Shintcovsk and colleagues demonstrated that nicotine had no influence on osteoclastogenesis [68]. Therefore, nicotine may not be a major compound responsible for impaired bone metabolism in bone cells.

Many studies have reported the negative relationship between CS and bone mass [69,70,71]. However, early diagnosis of impaired bone metabolism remains a great challenge. Due to the lack of obvious symptoms in the early stages of CS-induced osteoporosis, the bone metabolism of smokers has been impaired for a long period of time when they start to feel back pain or suffer from fractures [72,73]. Actually, after reaching peak bone mass around the age of 30 in adults, bone loss occurs at varying rates, and smoking significantly accelerates this process. Hence, it is highly desirable to detect the alternations of bone metabolism in the early stages to prevent the development of osteoporosis [4].

Several factors secreted into the blood by osteoblasts and osteoclasts regulate osteoblast-medicated bone formation and osteoclast-mediated bone resorption [74,75]. These serum markers represent a crucial complementary diagnostic tool to bone mineral density [76]. In general, monitoring these secreted markers could be helpful in determining the alternations in bone metabolism before the manifestation of osteoporosis. Figure 1 summarizes the direct effects of CS on bone metabolism.

#### 2.2.1. Markers for Bone Formation

Hydroxyproline (HYP) and type I collagen N- and C-terminal propeptides (PINP and PICP/CICP) in blood are commonly used to evaluate bone formation [77,78,79]. Ehnert and colleagues demonstrated that following total joint replacement, heavy smokers had a greater risk of prolonged hospital stay due to reduced levels of CICP as compared to nonsmokers [80]. Additionally, there are alternative markers involved in osteogenesis. Gürlek et al. reported that smoking significantly decreased salivary levels of osteocalcin (OC, later stage of osteogenic differentiation) [81]. Supporting these data, in a rat model, Gao et al. reported that chronic exposure to smoking significantly decreases levels of bone-specific alkaline phosphatase (BAP, early stage of osteogenic differentiation) and OC, resulting in osteoporotic bone [82].

#### 2.2.2. Markers for Bone Degradation

C-terminal telopeptide (CTX) and N-terminal telopeptide (NTX) in blood are commonly used to represent collagen-I degradation [78,79]. Oncken et al. demonstrated that smoking cessation contributed to a reduction in NTX levels in postmenopausal women [83]. In addition, there are alternative markers involved in osteoblastogensis. Our previous in vitro study revealed that chronic exposure to CSE-enhanced carbonic anhydrate II (CA II, early stage of osteoclastic differentiation [84]) and tartrate-resistant acid phosphatase 5b (TRAP 5b, later stage of osteoclastic differentiation [85]) activities in a bone co-culture system, resulting in an osteoporotic microenvironment [86].

Although these markers can represent bone metabolic alternations, the underlying mechanisms by which CS induces an osteoporotic bone environment are not yet fully understood. Understanding the regulatory mechanisms in bone cells impaired by CS will enable us to obtain insights into the development of CS-induced osteoporotic bones and screen relevant preventive treatment strategies for smokers with osteoporotic bones.

#### 2.2.3. RANKL-RANK-OPG Pathway

It has been shown that the receptor activator of nuclear factor-kappa B ligand (RANKL)-RANK-osteoprotegerin (OPG) (RANKL-RANK-OPG) pathway exerts a crucial role in osteoclast differentiation and bone resorption [87,88,89]. Leibbrandt and colleagues revealed that osteoclasts failed to be differentiated in a RANKL and RANK knockout mouse model, indicating RANKL is crucial for osteoclastic differentiation and activation [90]. RANKL, a membrane protein, is synthesized and released by osteoblasts, which is capable of binding to the RANK receptor located on osteoclasts, activating osteoclastogenesis [91]. As a decoy receptor for RANKL, OPG suppresses osteoclastogenesis by binding to RANKL [92]. Hence, the balance between RANKL and OPG exerts a vital role in regulating bone metabolism [93].

Currently, few studies have been conducted regarding the association between CS and the RNKL-RANK-OPG pathway. In terms of human studies, researchers have demonstrated that smoking significantly reduced OPG levels, leading to an increased RANKL/OPG ratio [45,94]. Consistently, in an in vitro bone co-culture system, our previous research showed that chronic exposure to CSE enhanced osteoclast function via upregulation of the RANKL/OPG ratio, resulting in an osteoporotic microenvironment, and bisphosphonates can be used to counteract the effects of CSE on the RANKL/OPG pathway [86].

Several studies have shown the RANKL-RANK-OPG pathway is also involved in indirect pathophysiological mechanisms (PTH-vitamin D axis and gonadal hormones), which makes this pathway of great therapeutic interest [90]. Moreover, this pathway, in interaction with several factors, such as prostaglandin E2 and interleukins, further affects bone metabolism [90]. For instance, estrogen and androgen could enhance OPG levels in osteoblasts and downregulate the RANKL/OPG ratio, further inhibiting osteoclast differentiation [95].

#### 2.2.4. Wnt/β-catenin Pathway

The canonical Wnt/β-catenin signaling pathway may be one potential mechanism that is involved in the regulation of RANKL/OPG balance. Glass et al. reported that the activation of the Wnt/β-catenin pathway in osteoblasts can downregulate the RANKL/OPG ratio, leading to compromised osteoclast differentiation and bone resorption [96]. Apart from suppressing osteoclastogenesis, it also enhances the osteogenic differentiation of MSCs [97], bone formation [98] and mineralization [99]. Therefore, the Wnt/β-catenin pathway appears to exert a dual role in bone remodeling, enhancing osteogenesis on the one hand and suppressing osteoclastogenesis on the other hand. Understanding and characterizing the Wnt/β-catenin pathway in detail is believed to be critical to guiding the interaction between osteoblast and osteoclast [100].

Wnt ligands, which secrete glycoproteins, are engaged in activating the Wnt/β-catenin canonical pathway [101]. When Wnt ligands (Wnt 3a and 5a) bind with receptor complexes, including Frizzled and LRP5/6, the Wnt/β-catenin pathway is activated to accumulate β-catenin in the cytoplasm of osteoblasts [102]. Then, the accumulated β-catenin transfers into the nucleus to regulate the transcription of osteogenic-related genes such as Osterix, *RUNX2* and *OPG* [103]. Eventually, the accumulated OPG binds with RANKL and prevents it from integrating with RANKL located on osteoclasts, inhibiting osteoclastogenesis [104]. Additionally, sclerostin and Dickkopf-1 (DKK1) are potent targets, blocking the canonical Wnt signaling pathway [105]. DKK1 is expressed by osteocytes and mature osteoblast, while sclerostin is expressed by osteocytes [106].

To date, few studies have been carried out showing the association between CS and inhibitors of Wnt signaling. Jorde et al. reported that smokers had higher levels of DKK1 and lower levels of bone formation (PINP) when compared to nonsmokers [39]. Supporting these data, Miranda et al. revealed that smoking could enhance the levels of DKK1 and sclerostin in patients with periodontitis [107]. Thus, the inhibition of Wnt signaling by CS may result in the upregulation of the RANKL/OPG ratio, leading to excessive osteoclast differentiation and bone resorption (Figure 2).

#### 2.2.5. Aryl Hydrocarbon Receptor (AhR) Pathway

Smoking exposure potently activates the cellular Aryl hydrocarbon receptor (AhR) [108], as cigarette smoke contains a large number of exogenous ligands for AhR, such as dioxin-like derivatives represented by 2,3,7,8-tetrachlorodibenzo-p-dioxin (TCDD) and a variety of polycyclic aromatic hydrocarbon (PAH)-based compounds such as benzo(a)pyrene (BaP) [109]. The AhR is one of the ligand-activated transcription factors for sensing environmental pollutants, which stays in an inactive form in the cytoplasm. Upon binding to exogenous ligands from smoke, the conformation of the AhR is changed, which enables AhR translocation into the nucleus, where it heterodimerizes with the aromatic hydrocarbon receptor nuclear transfer protein (ARNT) and activates xenobiotic response elements (XRE), to the initial transcription of target genes (CYP1a1/CYP1a2/CYP1b1) and leads to different toxicological and pathological effects [109,110]. Recent studies have shown that endogenous activation of the AhR has multiple roles in the regulation of bone remodeling and its associated signaling pathways, which may be an essential mechanism for smoking-induced osteoporosis [111].

Jameel et al. reported that BaP and TCDD in cigarette smoke caused AhR activation and a significant increase in CYP1A1 and CYP1A2 expression, resulting in enhanced OC differentiation and compromised bone mass in wild-type mice but not in AhR^-/-^ mice [33]. Moreover, TCDD administration was unable to stimulate osteoclastogenesis in bone marrow macrophages (BMMs) from CYP1A1/CYP1A2 double knockout and CYP1A1/CYP1A2/CYP1B1 triple knockout mice, unlike in BMM from wild-type mice [33]. These results suggest that AhR-induced CYP1 expression mediates at least in part the activation of osteoclastogenesis by AhR ligands from cigarette smoke. Further studies found that RANKL-stimulated osteoclastogenic signaling (e.g., phosphorylation of Akt, MAPK and NF-κB) was impaired in AhR^-/-^ BMMs [112]. Furthermore, the AhR was found to be activated by RANKL at an earlier stage than signature osteoclastic genes (e.g., CSTK and NFATc1) in pre-osteoclasts, suggesting a regulatory role of the AhR in OC differentiation. BaP was shown to induce higher levels of c-Fos in RANKL-stimulated BMMs, but c-Fos was not induced by BaP or RANKL in AhR^-/-^ BMMs [112]. Consistently, the AhR activation in BMMs was also found to lead to significant upregulation of c-Fos and NFATc1 expression, thereby initiating osteoclastic differentiation [113]. Taken together, the AhR pathway mediates excessive osteoclastic differentiation and bone resorption caused by smoking exposure, which could be a potential target for the prevention of smoking-induced osteoporosis (Figure 3).

## 3. Limitations and Future Overview

Studies on CS and bone metabolism are limited and have mostly been conducted in vitro. In vivo models of CS-induced osteoporosis still need further standardization. Therefore, the mechanisms of CS-induced osteoporosis that we have summarized in this review have not been fully validated by multiple studies. Comprehensive and systematic research is needed to reveal the key mechanisms of smoking-induced osteoporosis in the future. Bone metabolism is a process involving a variety of cells, including osteoblasts, osteoclasts and osteocytes, and it is not sufficient to study the effects of smoke on a single cell type. Similarly, the composition of cigarette smoke is very complex, so it is clear that studying the effects of one chemical component alone is incomplete. Therefore, in the future, we need to rely on in vitro co-culture systems and 3D cell culture systems of bone cells, as well as standardized animal models of smoking exposure to complete the mechanistic study of CS-induced osteoporosis. Moreover, exploring the genetic relationships of BMD with cigarette smoking status may provide insight into the novel genetic mechanisms of osteoporosis. A recent study found CS has the strongest correlation with human BMD compared to other common lifestyles and identified multiple candidate genes related to BMD and smoking, such as MAP1LC3B, a biomarker in the process of autophagy [114]. These findings could provide new directions for future research into the mechanisms, as well as new genetic targets for the prevention and treatment of CS-induced osteoporosis.

## 4. Conclusions

Smoking is highly addictive and significantly increases the rate of bone loss. The prevention and treatment of CS-induced osteoporosis deserve widespread attention. There are currently no effective interventions to specifically counteract CS-induced osteoporosis, owing to the fact that the specific mechanisms by which CS affects bone metabolism are uncertain. CS may affect bone metabolism through indirect pathways, such as affecting weight, hormone levels and oxidative stress levels. According to the current findings, CS can also induce osteoporosis through direct action on the skeletal system: (1) the RANKL-RANK-OPG pathway exerts a crucial role in the regulation of osteoclastogenesis in bone cells exposed to CS; (2) the Wnt signaling pathway may be the potential mechanism that directly engaged in osteogenic differentiation and indirectly participates in the RANKL-RANK-OPG pathway to regulate osteoclastic differentiation under CS exposure, and as a result, the activation of the Wnt signaling pathway may be a novel therapeutic approach for smokers with CS-induced osteoporosis; and (3) various components of cigarette smoke specifically activate the AhR of BMMs, which lead to excessive osteoclastic differentiation and bone resorption. The inhibition of the AhR pathway could be a promising strategy for CS-induced osteoporosis. Furthermore, there appears to be a very strong relationship between CS and genetic alterations associated with the development of osteoporosis, which is an exciting direction for future research into the mechanisms of CS-induced osteoporosis. This review summarizes the latest research findings on the mechanisms between CS and bone metabolism, with the aim of providing new targets and ideas for the prevention of CS-induced osteoporosis, as well as providing theoretical directions for further research in the future.

## Figures and Tables

**Figure 1 genes-13-00806-f001:**
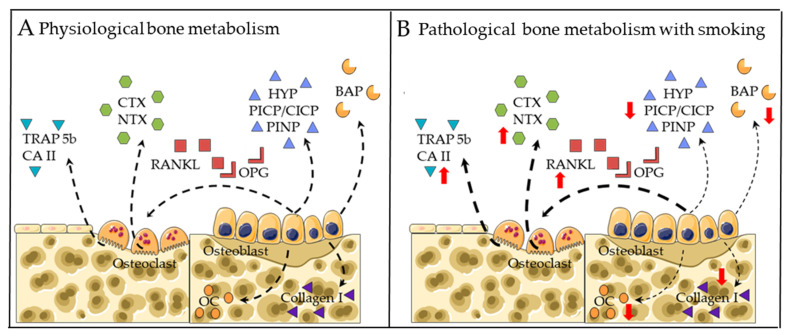
Established gene markers for bone metabolism in (**A**) physiological conditions and (**B**) CS exposure conditions. Bone resorption markers include: tartrate-resistant acid phosphatase isoform 5b (TRAP5b), carbonic anhydrate II (CA II), C-terminal telopeptide (CTX) and N-terminal telopeptide (NTX). Regulators of osteoclastogenesis include: receptor activator of nuclear factor kappa B ligand (RANKL) and osteoprotegerin (OPG). Bone formation markers include: osteocalcin (OC), collagen I, bone-specific alkaline phosphatase (BAP), hydroxyproline (HYP) and type I collagen N- and C-terminal propeptides (PINP and PICP/CICP). Dotted arrows represent expression. Red arrows represent altered expression (up or down).

**Figure 2 genes-13-00806-f002:**
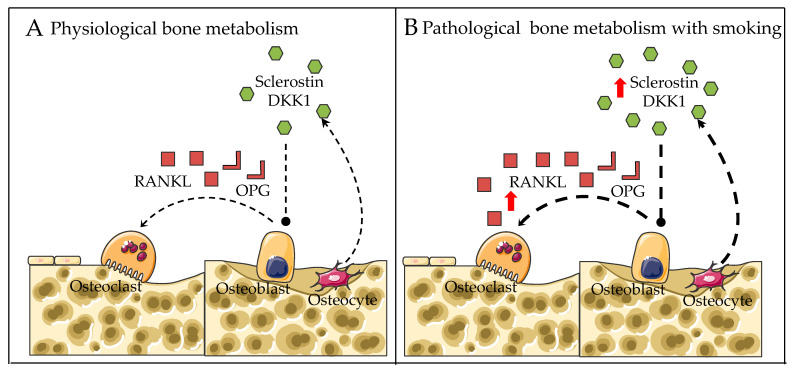
Proposed regulatory mechanisms for bone metabolism in (**A**) physiological conditions and (**B**) smoking exposure conditions. Dotted arrows represent expression. Red arrows represent altered expression (up or down).

**Figure 3 genes-13-00806-f003:**
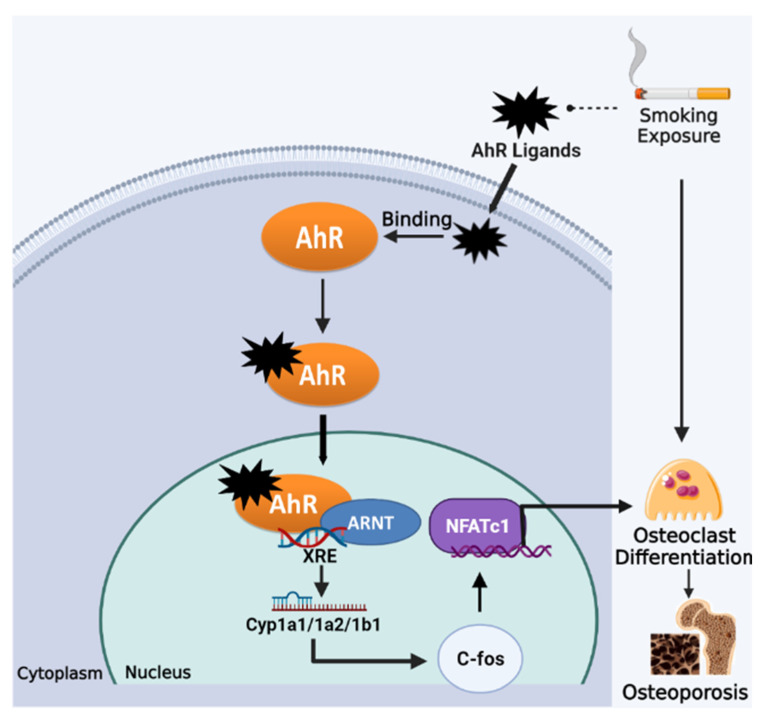
Smoking exposure induces osteoclast differentiation through the AhR pathway, leading to osteoporosis. Dotted lines represent release, arrows represent promotion.

**Table 2 genes-13-00806-t002:** Animal models of cigarette smoking-induced osteoporosis.

Animal	Smoking Mode	Intervention Dose	Intervention Duration	Result	Reference
Rat	Cigarette smoke inhalation	1 h each time twice a day	6 days per week for 16 weeks	Bone loss and decreased BMD	Zhuang et al. [29]
Rat	Cigarette smoke inhalation	2 h each day	9 weeks	Delayed fracture healing	Chang et al. [30]
Rat	Cigarette smoke inhalation	30 min each time twice a day	5 days per week for 6 months	Bone loss and decreased BMD	Levi et al. [31]
Mice	Cigarette Smoke inhalation	50 min each time twice a day	5 days per week for 24 weeks	Bone loss and decreased BMD	Xiong et al. [32]
Mice	Gavaged orally	BaP (120 mg/kg)	6 days	Bone loss and decreased BMD	Iqbal et al. [33]

## Data Availability

Not applicable.

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
