# Peer review of "An Overlooked Bone Metabolic Disorder: Cigarette Smoking-Induced Osteoporosis"

_genes, 2022, doi:10.3390/genes13050806_

Round 1

Reviewer 1 Report

This is a well written review on cigarette smoking and bone loss/osteoporosis. The authors summarized the recent advances in different signaling pathways involved in osteoblastogenesis and osteoclastogenesis that were affected by cigarette smoking.  The schematic illustration is very useful for summarizing key findings in this research direction. Following are few minor comments:

  1. The title needs to be improved. “Easily”  should be removed.
  2. The meaning of this sentence is not clear. “Due to the lack of obvious symptoms in the early stages of CS-induced osteoporosis, bone metabolism of smokers has been impaired for a long period of time when they have significant bone loss or already suffered back pain or fracture [60,61]”. Need rephase.
  3. Line 181, representing should be “regarding”.
  4. Line 205 “to guiding” should be “to guide”.
  5. Lines 208, 212, 213 “Integrate” should be “bind”. Integrate is not a common word for ligand and receptor interactions.
  6. Line 214, sclerostin and DKK1 are known Wnt-signaling inhibitors, not potential targets, if you want to say strong, it should be “potent”, not potential.
  7. Line 233 “Based like BaP” meaning not clear. May be “such as” or other appropriate wording to express what you want to say.
  8. References 39, 41, 82 have issues, no journal name. Are these books? Formats of references appears different.

Reviewer 2 Report

The authors should consider the followings:

  1. The authors should tabulate the relevant clinical studies (the authors should categorize the tables, by Phases in Clinical Studies, and by the number of recruited subjects, and by the primary outcomes) of cigarette smoking-induced osteoporosis.
  2. The authors should elaborate for the limitation(s) of the current review.
  3. The authors may further add section to the influences of e-cigarette to the topic.
  4. In introduction, the authors may specify the estimated epidemiology of the cigarette smoking-induced osteoporosis populations.
  5. In introduction, please briefly classify and standardize the grading of smokers, in terms of heavy smokers, and long-term smokers. i.e. How long or how heavy would it classify as heavy smokers, and long-term smokers.
  6. Please add a section of, future direction to gather the future plans from the authors.
  7. Please summarize the current therapies of cigarette smoking-induced osteoporosis, if any.
  8. The authors may use an informative table to summarize the relevant animal studies of the subject matter.
  9. The authors may improve the figure resolution of Figure 3.
  10. The authors should seek English professional for the Use of English in the article.

Reviewer 3 Report

This is a nice summary of existing literature on the possible biological signaling mechanisms influenced by cigarette smoke, that leads to bone metabolic disorders, and osteoporosis. What is lacking in the review in its present form is a discussion on possible genetic susceptibilities towards bone metabolic disorders and osteoporosis, and if/how cigarette smoke could exacerbate the problem. In addition, the Conclusion should provide a more comprehensive summary statement of the factors critical for healthy bone metabolism, and how cigarette smoking contributes to the dysregulation of those pathways. There are some typographical, and grammatical errors in the manuscript which should be corrected.
